# Gaming Behaviors and the Association with Sleep Duration, Social Jetlag, and Difficulties Falling Asleep among Norwegian Adolescents

**DOI:** 10.3390/ijerph19031765

**Published:** 2022-02-04

**Authors:** Regina Hamre, Otto Robert Frans Smith, Oddrun Samdal, Ellen Haug

**Affiliations:** 1Department of Health Promotion and Development, University of Bergen, 5020 Bergen, Norway; regina.hamre@idrettsklyngevest.no (R.H.); Oddrun.Samdal@uib.no (O.S.); 2Department of Health Promotion, Norwegian Institute of Public Health, 5015 Bergen, Norway; robert.smith@fhi.no; 3Department of Teacher Education, NLA University College, Bergen, Pb 74 Sandviken, 5812 Bergen, Norway

**Keywords:** sleep duration, weekday sleep, weekend sleep, social jetlag, difficulties falling asleep, gaming behaviors, problem gaming, gaming addiction, engaged gaming, adolescents

## Abstract

The relationship between gaming and sleep is mostly informed by studies of addictive gaming behavior, thus limiting our understanding of sleep in the context of nonproblematic engaged gaming. The present study investigated whether addicted, problem, and engaged gaming behavior was associated with sleep duration, social jetlag, and difficulties falling asleep. The sample consisted of 13- and 16-year-old Norwegian adolescents (n = 3228) participating in the Health Behavior in School-Aged Children (HBSC) survey in 2018. Participants were categorized into addicted, problem, engaged, and normal/non-gaming behavior groups according to which GAS-7 criteria they fulfilled. Robust generalized linear mixed models with a random intercept for class ID were used to examine the association between the sleep variables and gaming behavior. Addicted gaming behavior was unfavorably associated with all sleep parameters. The findings for engaged gaming and problem gaming behavior were somewhat mixed. Engaged gamers slept less on weekends, less on weekdays for those aged 16, and experienced greater social jetlag compared to the normal/non-gaming group. Problem gamers experienced greater social jetlag and had higher odds of experiencing difficulties falling asleep. Overall, the results suggest that all types of gaming behaviors might harm sleep health, but to a greater extent for the addicted gamers.

## 1. Introduction

Games are a natural part of human life. As an extension of play, games have presented entertaining opportunities to engage in and improve cognitive abilities such as planning, strategy, memory, communication, and cooperation [1]. However, modern video games are surrounded by concerns for players’ health and wellbeing, especially when gaming takes precedence over necessary activities, leading to negative consequences [2]. This problematic pattern of gaming is now included in ICD 11 as “gaming disorder” [3]. Accordingly, the WHO encourages all gamers to be alert to the amount of time they spend on games, particularly when it is to the exclusion of other daily activities [2].

One of the activities gaming can affect is sleep. Sleep is an essential health behavior that sustains human wellbeing [4]. Sleep is also a complex process where good sleep encompasses an appropriate timing, amount, and quality [5]. Recent evidence in the European context shows that youth sleep is short and ill-timed, and a large proportion of adolescents have difficulties falling asleep [6,7,8,9]. This puts a substantial number of youths at risk for mood and mental disorders [10,11,12], negative health outcomes such as obesity [13,14], and decreased academic performance [15].

Furthermore, many teenagers are experiencing social jetlag [6,16]: a discrepancy between the internal body clock and social time, causing large differences in weekday and weekend sleep timing [17]. This phenomenon is especially relevant to the teenage population where biological shifts towards eveningness put them in conflict with societal schedules such as school start times [18]. Social jetlag is measured as the difference between weekday sleep (corresponding to social time) and weekend sleep (corresponding to biological time). However, there is no agreed upon measurement point. The original formula measures differences in mid-sleep [17], but formulas using bedtime have also been proposed [19]. Emerging evidence suggests that social jetlag may be associated with lower work and school performance [20,21], metabolic risk factors, BMI, and eating habits [21,22,23].

Several mechanisms by which screen-based technology can affect sleep have been proposed [24]. Time spent on screen-based activities can directly displace sleep or sleep-aiding behaviors such as winding down to sleep both before and after going to bed [9,25] as well as displacing physical activity [26,27]. Furthermore, light emitted by devices affects sleep regulatory mechanisms such as melatonin [18,28,29,30,31,32]. The engaging nature of screen-based activities can also cause physical or psychological arousal [33,34].

Both screen time and gaming behavior are associated with adverse effects on sleep health. Recent reviews have demonstrated a consistent association between screen time and sleep problems [35,36]. Furthermore, a trend study of sleeping difficulties among European adolescents found that screen time was significantly associated with problems falling asleep, and the relationship has strengthened over the period from 2002 to 2014 [7]. A paucity of studies has also found that screen time affects social jetlag measured as differences in weekday–weekend mid-sleep and “wake lag”, measured as differences in weekday–weekend rise times [37,38]. However, no studies have looked at technology use and social jetlag defined as a difference in bedtime on weekdays and weekends [19]. Moreover, a recent literature review found that problematic gaming was associated with shorter sleep duration, increased likelihood of reporting poor sleep quality, daytime sleepiness, and sleep problems [39].

Notwithstanding the increasing research in the field, the conceptualization of problematic gaming and its measurements remain disputed [40,41]. The inclusion of gaming disorder in the ICD 11 has not surmounted the fact that there is no agreed-upon tool to assess problematic gaming [42]. Problematic gaming has largely been viewed as a behavioral addiction and classified based on the presence of addiction components such as salience, conflict, loss of control, relief/mood modification, tolerance, withdrawals, and relapse [43]. However, it has been increasingly acknowledged that scales solely focused on addicted gaming behavior might miss or misdiagnose other types of behaviors, especially if polythetic scoring is used where only some of the criteria need to be endorsed [42,44,45].

Later analysis of addiction components has shown that rather than measuring the concept of problematic gaming, the components instead represent two stable factors: pathological addicted gaming and nonpathological engagement [46,47]. Based on this, Charlton and Danforth [46,47] proposed a distinction between core components that relate to addicted gaming (abstinence, relapse, and loss of control), and peripheral components relating to engaged gaming (salience, tolerance, and relief). Thus, being preoccupied with gaming, increasing use over time, and experiencing pleasant emotional states may reflect a normal engagement behavior, and not necessarily addiction.

This two-factor solution has also been found to be valid for the seven-item version of the “Gaming addiction Scale for Adolescents” (GAS-7) [45,48]. In this scale, specially developed for youth, the core criteria “problems” has been added to Browns [43] addiction components [45,48]. This criterion refers to problems of excessive game play, caused by the activity taking precedence over other necessary activities such as school, work, and socializing [45]. In a recent confirmatory factor analysis, a two-factor structure of the GAS-7, which separates peripheral criteria from core criteria, showed a better fit compared to the original one-factor solution in a Norwegian population [48]. In sum, there are empirical and theoretical grounds warranting a separation of core and peripheral components to better capture gaming behavior.

The distinction between addiction and engagement is important. Only focusing on the consequences of addicted gaming greatly limits the understanding of gaming behaviors and possible health impacts. A study categorizing participants using the GAS-7 as addicted gamers (participants fulfilling all four core criteria), problem gamers (participants fulfilling 2–3 core criteria), engaged gamers (participants fulfilling all peripheral criteria and none or one of the core criteria), and normal/non-gamers (all other participants), found differential associations with subjective health complaints [49]. Addicted gamers and problem gamers had a significantly greater risk of experiencing subjective health complaints compared to the contrast group of normal/non-gamers. However, the engaged gamers did not differ from the contrast group. One of the subjective health complaints was difficulties falling asleep. Addicted and engaged gamers spent comparable time on gaming, putting both groups at risk for sleep problems. However, only addicted and problem gamers had a greater risk of difficulties falling asleep. Taken together, the studies suggest that, although both addicted and engaged gamers can experience adverse sleep effects considering the time they spend on games, this may not be the case for engaged gaming behavior.

This study applied the categorization of GAS-7 into addicted, problem, or engaged gaming behavior [49] and investigated the association with sleep duration, social jetlag, and difficulties falling asleep among 13- and 16-year-old Norwegian adolescents. Specifically, we anticipated that addicted and problem gaming was negatively associated with sleep parameters, but we were more agnostic about the relationship between engaged gaming behavior and sleep, given the mixed findings in the literature. Age, gender [6,7,16,50,51], socioeconomic status [6,51], and physical activity [7,27] are known to be related to sleep and gaming and were thus adjusted for as possible confounders. Age was also included as a possible moderator, as age greatly influences sleep in the adolescent years and is also theorized to influence gaming habits [18,24]. We anticipated that any relationships between gaming behaviors and sleep would strengthen with increasing age.

## 2. Methods

### 2.1. Design and Data Collection

The present study is based on data from the Norwegian sample of the cross-national Health Behavior in School-aged Children survey (HBSC), a WHO study conducted in more than 50 countries and regions every four years [52]. The data was collected in 2018 among a national sample of 11-, 13-, 15-, and 16-year-olds, where the sampling unit was school classes. The study was conducted in accordance with the declaration of Helsinki, and the protocol was considered and recommended by the Norwegian Centre for Research Data (id code 56623/4/LH), which is the Norwegian national authority for data protection. All subjects (participants or parents of participants under the age of 16) gave their informed consent. All students were also informed on the day of the survey that participation was voluntary and that they could withdraw their consent at any point regardless of a previously given consent. The survey was completed digitally during school in a 45 min session after receiving standardized instructions from the teacher. On class level, the response rate was 78% among the 11-, 13-, and 15-year-olds and 87% among the 16-year-olds (upper secondary school). The current study uses data from the 13-year-olds (n = 1024) and 16-year-olds (n = 2204), where 49.8% were boys and 50.2% girls

### 2.2. Measures

#### 2.2.1. Gaming Behavior

Gaming behavior was measured with scores on the GAS-7 scale, consisting of 7 items assessing the components salience, conflict, mood modification, tolerance, withdrawals, relapse, and problems [45]. The responses were 1 = never, 2 = almost never, 3 = sometimes, 4 = often, and 5 = very often. A component was considered as endorsed if the participant answered 3 or higher [45], in line with the study of Brunborg et al. [49]. The participants were split into four different gaming behavior groups according to which criteria they fulfilled. If participants fulfilled the core criteria abstinence, relapse, loss of control, and problems, they were assigned to the ‘addicted gaming’ group. Participants fulfilling 2–3 of these criteria were assigned to a ‘problem gaming’ group. Participants fulfilling the peripheral criteria salience, tolerance, mood modification, and one or none of the core criteria were assigned to the ‘engaged gaming’ group. Participants that could not be categorized as addicted, problem, or engaged gamers were assigned to a contrast group of normal/non-gamers. Thus, belonging to the contrast group does not entail the absence of a gaming behavior altogether, but only the absence of addicted, problem and engaged gaming behavior.

#### 2.2.2. Sleep Variables

Sleep duration was measured by asking when participants usually go to bed and get up during weekdays and weekends. The responses were listed in half-hour intervals. For weekdays, bedtimes ranged from “21:00” to “02:00 or later”, and rise time ranged from “no later than 05:00” to “08:00 or later”. Weekend bedtimes ranged from “no later than 21:00” to “04:00 or later” and rise times from “no later than 07:00” to “14:00 or later”. The differences between bedtime and rise time were calculated and subtracted from 24 h to find sleep duration.

Social jetlag was calculated as the difference between weekday and weekend bedtime. Bedtime was used as opposed to the formula in Wittmann et al. [17] that uses mid-sleep, as the original formula can measure sleep debt in some participants [19]. Positive values of social jetlag indicate a later bedtime on weekends compared to weekdays.

Difficulties falling asleep is an item derived from the subjective health complaints scale applied in the HBSC study [52,53]. The participants were asked how often they had difficulties sleeping during the past 6 months. The responses were 1 = almost every day, 2 = more than once a week, 3 = almost every week, 4 = almost every month, 5 = rarely or never. The variable was dichotomized with a cut-off at 2 to capture weekly difficulties sleeping.

#### 2.2.3. Socioeconomic Status

Socioeconomic status (SES) was measured by a summary score on the family affluence scale (FAS), a scale that measures a material dimension of SES derived from the characteristics of the family’s household that consists of six items (family car, number of computers, own bedroom, family holidays, number of bathrooms, and dishwasher in home) [54]. The summary score was split into low (upper 20%), middle (middle 60%), and high (upper 20%) SES for the descriptive analysis.

#### 2.2.4. Physical Activity

Physical activity of moderate to vigorous intensity (MVPA) was assessed with one item. The participants were asked to report the number of days over the past week during which they were physically active for a total of at least 60 min. The question was introduced by a text defining moderate-to-vigorous physical activity as any activity that increases the heart rate and makes the person out of breath some of the time, with examples provided [52,55]. The score was dichotomized into four days or less = 0, and 5 days or more = 1.

### 2.3. Statistics

Cases with missing values on variables of interest were dropped from the analyses. The frequency of missing values was 13.7% for the gaming variables and 9.0% for the physical activity variable. All other variables had frequencies of missing values below 5%. In the unadjusted regression analyses, about 14% of the cases were excluded due to missingness, while this was 22% for the adjusted regression analyses. We did not consider alternative methods to deal with missingness (e.g., multiple imputation) as we were not able to identify variables with complete data that correlated sufficiently high enough with the gaming and physical activity variables to substantially alter the estimates based on listwise deletion [56].

Robust generalized linear mixed models with a random intercept for class ID were used to examine the association between the sleep variables and gaming behavior. Sleep duration and social jetlag were modelled as continuous variables (normal distribution, identity link function), whereas difficulties falling asleep was modelled as a binary variable (binomial distribution, logit link function). The gaming behaviors were dummy coded with the normal/nongaming behavior set as the reference category. Thus, the results from the analysis can be interpreted as the difference in scores from the nongaming/normal gamer category and the other categories of gaming behavior. In previous research both sleep and gaming have shown associations with physical activity, SES, age, and gender [6,7,16,27,50,51]. Therefore, these variables were included as possible confounders and adjusted for in the models. Furthermore, age was tested as a possible moderator [18,24]. The statistical analyses were conducted using IBM SPSS Statistics version 26.0.

## 3. Results

### 3.1. Baseline Characteristics

Table 1 shows the characteristics of the sample. A quarter of the sample could be classified as addicted, problem, or engaged gamers, with problem gamers being the most frequent. The average sleep duration was 7 h 36 min on weekdays, and 9 h 38 min on weekends, with bedtimes on weekend 1 h 53 min later than on weekdays. Nearly a quarter of the sample had difficulties falling asleep more than once a week.

### 3.2. Association between Gaming Behavior and Sleep Duration in Weekdays and Weekends

Table 2 and Table 3 show the associations between gaming behaviors and sleep duration on weekdays and weekends. On weekdays, only addicted gaming behavior was negatively associated with sleep duration. For engaged gamers, a moderating effect of age on the relationship was present, where a negative association emerged with increasing age (Figure 1). On weekends, both addicted and engaged gaming behavior was negatively associated with sleep duration. No interaction effect of age was observed for the association of gaming behavior and weekend sleep.

Contrast group set as reference. B (unstandardized regression coefficient) = hours. Bold writing indicates statistical significance at the *p* < 0.05 level.

### 3.3. Associations between Gaming Behaviors, Social Jetlag, and Problems Falling Asleep

Table 4 shows that both addicted and engaged gaming behavior was associated with greater social jetlag when adjusting for covariates. Belonging to the problematic gaming behavior group was associated with greater odds of having difficulties falling asleep more than once a week (Table 5). No interaction effect of age was observed for the association of gaming behaviors and social jetlag, nor difficulties falling asleep.

## 4. Discussion

This study aimed to investigate if addicted, problem, or engaged gaming behavior was associated with sleep duration, social jetlag, and difficulties falling asleep among 13- and 16-year-old Norwegian adolescents. Specifically, we anticipated that addicted and problem gaming was negatively associated with sleep parameters. Still, we were more agnostic about the relationship between engaged gaming behavior and sleep, given the mixed findings in the literature. Any existing relationships were hypothesized to strengthen with increasing age. The results showed that addicted gaming behavior was unfavorably associated with all sleep parameters, confirming our hypothesis. However, the findings for problem gaming were somewhat mixed. Problem gamers experienced greater social jetlag and had higher odds of experiencing difficulties falling asleep. No moderation of age was present for addicted and problem gamers, thus the hypothesis of moderation by age was not supported for these groups. Engaged gamers slept less on weekends, and less on weekdays for those aged 16, partially confirming our hypothesis, and experienced greater social jetlag compared to the non-gaming/normal group.

The negative associations between addicted gaming behavior and sleep duration are largely in line with previous findings [39,57,58]. In the present study, addicted gamers slept approximately half an hour less on weekdays compared to the contrast group. This is slightly higher than the estimate of 21 minutes reported in a recent meta-analysis of problematic gaming and sleep [39]. Conversely, some studies have found substantially larger differences in sleep duration: Hawi et al. [58] found that Lebanese youth with internet gaming disorder (IGD 20) slept 2 h less on weekdays compared to casual gamers and gamers with a risk of developing disordered gaming behavior. However, other studies have failed to find a significant or meaningful association between gaming behaviors and sleep [59,60], though the studies that failed to find a relationship did not differentiate between weekday and weekend sleep. Further, participants reported average hours of sleep, which calls for mentally averaging weekday and weekend sleep.

Except for the addicted gamers, only the 16-year-old engaged gamers slept less on weekdays compared to the contrast group. This moderation effect of age is interesting, as a large Norwegian study using the same scale and categorization of gamers found that both the engaged and the problem behavior groups had increased screen-time use compared to non-gamers/normal gamers, well exceeding an average of 2 h per day [49]. In a study by Hysing et al. [61] all screen time surpassing 2 h on a single device was negatively related to sleep duration in a dose–response relationship. In comparison to our findings, Hawi et al. [58] failed to find an association between sleep duration and youth fulfilling some but not all of the IGD 20 criteria. The effect of gaming on sleep at age 16 is lamentable, as older adolescents’ struggle the most with sleep. Adolescents’ sleep health typically declines with increasing age due to biological changes, decrease in parental monitoring, and increases in social demands such as schoolwork, after-school-jobs, and social life [18].

The effect of moderation by age found amongst the engaged gamers is in line with Cain and Gradisar’s [24] theory of electronic media and sleep: The autonomy adolescents gain with age can affect one’s gaming habits, which in turn can affect sleep. Thus, one possible explanation for the lack of association among 13-year-old engaged gamers is that parents are more involved and enforce regulations concerning sleep, protecting against the adverse effects of technology use. Indeed Pyper et al. [62] found that parents enforcing bedtime regulations were 1.59 times more likely to have a child that fulfilled sleep recommendations. Furthermore, parents enforcing this protective behavior decreased from 13-year-olds to 16-year-olds simultaneously as the proportions of adolescents meeting sleep recommendations decreased [62]. This implies that the autonomy gained during the adolescent years might create more opportunities for engaged gaming habits to displace sleep.

The effects of gaming might vary across the week. Still, few studies have differentiated weekend and weekday sleep when examining the relationship between gaming behavior and sleep. The current study identified that both engaged and addicted gaming was notably associated with less sleep on weekends, breaking from the persistent teenage tendency to offset sleep debt from previous weekdays [18]. The more evident association between engaged gaming and sleep duration on weekends might be explained by gamers trying to allocate gaming to the weekend as a conscious effort to minimize disrupting effects in everyday life. Indeed, Hawi et al. [58] found that nonaddicted gamers doubled the time spent on games during the weekends, whereas addicted gamers had comparable screen time on weekdays and weekends. Furthermore, experimental research has found that the length of the gaming session matters irrespective of bedtime. In King et al. [33] shorter gaming sessions (50 min) did not affect sleep duration, whereas longer sessions did (150 min). Taken together, the habit of allocating gaming to the weekends might make engaged players vulnerable to both direct displacement of sleep and activating effects delaying sleep onset.

Both addicted and engaged gamers had larger social jetlag compared to the contrast group. This is a novel finding; no study has, to our knowledge, specifically examined gaming behavior and social jetlag. However, broader categories of time spent on sedentary screen behaviors have been investigated. Hena and Garmy [37] found that screen time over 4 h was associated with larger social jetlag, an amount that the average European teenager has exceeded since 2002 [7]. For engaged gamers in particular, the social jetlag might be a consequence of gaming displacing sleep to a greater degree on weekends, in alignment with the observed decreased sleep duration in the weekends.

Addicted and problematic gaming behaviors were also associated with higher odds of having difficulties falling asleep. Using the same measurement of sleep difficulties as the present study, Brunborg et al [49] and Männikkö et al. [63] found a significant association between addicted gaming and difficulties falling asleep. Further, Rehbein et al. [64] found that IGD was associated with a higher occurrence of problems sleeping in the past week. Similarly, significant associations between addicted gaming, sleep onset length [65], sleep quality [66,67], and insomnia [68,69,70] have also been found. As with weekday sleep, the missing association with engaged gaming in the present study is surprising, as screen time alone has been associated with difficulties falling asleep [7,36,61], suggesting an association with non-problematic gaming.

Several underlying mechanisms of the relationship between technology use and sleep have been proposed, such as displacement of bedtime [71,72,73]. This displacement effect could result from games inducing a state of “flow”. Flow is a sense of pleasant and positive immersion in an activity to the degree that one pursues the activity for its own sake; it becomes autotelic [74]. Flow is highly relevant concerning gaming behavior and sleep. It elicits distorted time perception, where activities become difficult to track, possibly leading to displaced sleep or sleep-aiding activities [71,72,73,74]. Indeed, greater flow states are associated with later bedtimes directly, and through time spent gaming [75]. Several conditions of flow-states are key parts of video games: clear goals, immediate feedback, matching of skill and challenge, high concentration demands, and a sense of escapism. Thus, flow states can be an underlying component of gaming that makes both engaged and addicted gamers vulnerable to displacement effects.

Further, emitted light and activation from gaming can make it more difficult to fall asleep [18,28,29,30,31,32,33,34], but the effect seems to be related to the length of exposure. In King et al. [33] prolonged exciting gameplay reduced sleep duration by 30 min compared to a control group that played for a shorter duration, even though bedtime was kept the same for both groups. Similarly, Wood et al [28] found that melatonin was suppressed at two hours of light exposure, but not one. Thus, the impact of light and activation might be greater during the weekend, perhaps due to longer gaming sessions, less external (parental) monitoring, and less incentive to self-monitor, giving rise to social jetlag. Staying up to play may also be a socially rewarding experience for many gamers. Most popular video games today (e.g., Fortnite, Counter Strike, World of Warcraft, and Minecraft when played on communal servers) are inherently social, where gamers communicate with each other in-game and trough external platforms such as Discord. Logging off to sleep may trigger disapproval from peers and fear of missing out. This pressure may also vary throughout the week, where the norm of staying up is stronger during the weekend compared to weekdays.

Taken together, our results suggest that engaged gamers seem to have a higher degree of control over their gaming, resulting in less sleep-disturbing effects compared to addicted gamers. For example, engaged gamers might employ more successful control measures, such as allocating gaming to the weekends, thus avoiding some of the effects on sleep observed among addicted gamers. This is underpinned by the underlying core criteria of the behaviors, particularly loss of control and problems. Both criteria are closely related to addiction behavior and not necessarily to engagement [41,64]. In this regard, it is interesting that the participants categorized as problem gamers did not experience the same degree of sleep problems, as they experienced 2–3 of the core criteria.

The overall findings show that both pathological and nonpathological gaming can be associated with adverse sleep outcomes. This may in part be a result of the engaging and social nature of video games. The relationships must be interpreted in the context of teenage sleep that is already short and ill-timed, and a large proportion of adolescents have difficulties falling asleep [6,7,8,9,16]. Thus, both addicted and engaged gamers, and to a lesser degree problem gamers, are at risk of experiencing adverse psychological, physical, and social outcomes [10,11,12,13,14,15,20,21,22,23]. Overall, the results should raise concern as youths spend increasing amounts of time gaming, a trend that seems to have accelerated during the COVID-19 pandemic [76,77,78,79,80].

### 4.1. Strengths and Limitations

This study adds to the literature on gaming and sleep by providing a nuanced perspective of different gaming behaviors in a large nationally representative sample of adolescents conducted with comprehensive methodological procedures. Additionally, the gaming behavior instrument has shown satisfying validity in the Norwegian population, outperforming the original GAS-7 one-factor solution [48]. However, the study also has limitations. Firstly, it is limited by its cross-sectional design where the direction of the relationship cannot be determined, and causal inferences cannot be made. Further, the study also relies on subjects’ recall. Previous studies have shown that subjective sleep measures are susceptible to underreporting [81]. Further, different subjects might have different perceptions of what bedtime entails. Bedtime is not the same as shuteye time [25]. A nuanced measure could have provided a more precise bedtime assessment. The study lacked measures of other screen-based actives that may be possible confounders in the relationship. Lastly, we cannot exclude the possibility for a certain degree of bias in the presented regression estimates as a result of moderate missing data rates for some of the included variables in this paper.

### 4.2. Implications and Directions for Future Research

The result from the current study demonstrates the usefulness of a differentiation between engaged gamers, problem gamers, and addicted gamers to obtain a nuanced understanding of the relationship with sleep. Thus, the findings add to the already existing research suggesting that gaming may harm teenage sleep. Although gaming and technology use is of importance to youth sleep, policy makers should be conscious of the disparity between societal factors such as school start times and circadian shifts in adolescents [18]. It is of paramount importance that treatment of problematic gaming becomes accessible to all individuals in need. For engaged gamers, the development of more organized in-school and after-school gaming activities (such as e-sports) may aid youths in better structuring the activity and reduce the need to play close to bedtime. This would also create meaningful social contexts where gamers can be educated on how to balance gaming with health behaviors.

Future research should, when possible, be wary of the distinction between addicted and other types of gaming behaviors to accurately measure and research the consequences of these gaming behaviors. Further investigation into the reliability and validity of the two-factor solution is warranted. Additional studies on technology and sleep would benefit from including more and diverse measures of the screen-based activities, such as the use of other technology, the duration of screen-time activities, as well as the inclusion of diverse sleep measures preferably separating bedtime from shuteye time.

## 5. Conclusions

Overall, the results from the current study suggest that both nonpathological (engaged) and pathological gaming behavior (addicted) might harm sleep health. These results demonstrate the usefulness of differentiation into different groups of gaming behaviors. The findings add to the already existing research on gaming as a correlate of teenage sleep. Addicted gaming behavior seems to be most unfavorably associated with important sleep parameters. However, both addicted and engaged gamers notably slept less over the weekend. Further, the gaming groups make up a relatively large proportion of the adolescent population, therefore the results raise concerns, as teenage sleep is already insufficient, with multiple reports of difficulties falling asleep.

## Figures and Tables

**Figure 1 ijerph-19-01765-f001:**
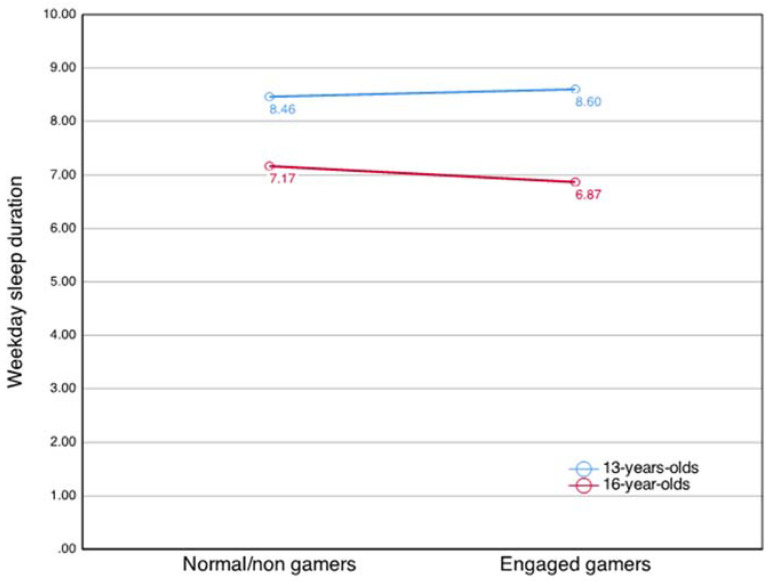
Effect of age on the association of engaged gaming and weekday sleep duration.

**Table 1 ijerph-19-01765-t001:** Baseline characteristics.

Variable	Estimate
Female gender % (n)	50.2 (1614)
Age % (n)	
13-year olds	31.7 (1024)
16-year olds	68.3 (2204)
Family Affluence % (n)	
Low	20.6 (646)
Middle	62.4 (1957)
High	17.0 (534)
MVPA ≥ 5 days % (n)	36.3 (1174)
Family affluence (SD)	7.79 (1.80)
Gaming behavior % (n)	
Addicted	4.40 (123)
Problem	14.9 (416)
Engaged	6.40 (179)
Normal/nongamers	74.2 (2069)
Sleep measures	
Weekday sleep duration (SD)	7.83 (1.17)
Weekend sleep duration (SD)	9.64 (1.36)
Social Jetlag (SD) *	1.88 (1.13)
Weekly sleeping difficulties % (n)	23.2 (774)

* Positive values of social jetlag indicate later bedtimes on weekends compared to weekdays.

**Table 2 ijerph-19-01765-t002:** Crude and adjusted associations of gaming behavior and sleep duration on weekdays.

	Model 1			Model 2			Model 3		
	b	CI 95%	*p*	b	CI 95%	*p*	b	CI 95%	*p*
Addicted	**−0.48**	**[−0.71, −0.25]**	**0.000**	**−0.56**	**[−0.80, −0.31]**	**0.000**	−0.40	[−0.84, −0.46]	0.079
Problem	−0.07	[−0.20, 0.06]	0.293	−0.09	[−0.21, 0.04]	0.172	0.26	[−0.16, 0.21]	0.788
Engaged	−0.15	[−0.35, 0.06]	0.161	−0.13	[−0.34, 0.08]	0.217	0.15	[−0.12, 0.42]	0.284
Age 16				**−1.15**	**[−1.28, −1.02]**	**0.000**	**−1.08**	**[−1.21, −0.95]**	**0.000**
Gender female				−0.07	[−0.18, 0.03]	0.151	**−0.08**	**[−0.19, 0.17]**	**0.033**
SES				0.01	[−0.03, 0.02]	0.562	−0.01	[−0.04, 0.02]	0.532
MVPA ≥ 5 days				**0.09**	**[−0.19, −0.01]**	**0.046**	**−0.10**	**[0.19, 0.01]**	**0.038**
Addicted × age							−0.23	[−0.76, −0.30]	0.390
Problem × age							−0.19	[−0.25, −0.02]	0.133
Engaged × age							**−0.45**	**[−0.82, −0.06]**	**0.022**
Random intercept	**80.02**	**[7.91, 8.13]**	**0.000**	**8.84**	**[8.60, 90.07]**	**0.000**	**8.79**	**[8.55, 90.03]**	**0.000**

Contrast group set as reference. B (unstandardized regression coefficient) = hours. Bold writing indicates statistical significance at the *p* < 0.05 level.

**Table 3 ijerph-19-01765-t003:** Crude and adjusted associations of gaming behavior and sleep duration on weekends.

	Model 1			Model 2			Model 3		
	b	CI 95%	*p*	b	CI 95%	*p*	b	CI 95%	*p*
Addicted	**−0.42**	**[−0.72, −0.13]**	**0.004**	**−0.40**	**[−0.69, −0.11]**	**0.008**	**−0.48**	**[−0.94, −0.01]**	**0.045**
Problem	−0.10	[−0.25, 0.05]	0.176	−0.08	[−0.24, 0.06]	0.332	−0.12	[−0.32, 0.09]	0.281
Engaged	−0.24	[−0.49, 0.00]	0.054	**−0.32**	**[−0.58, −0.05]**	**0.019**	**−0.50**	**[−0.92, −0.09]**	**0.016**
Age 16				**−0.51**	**[−0.62, 0.40]**	**0.000**	**−0.54**	**[−0.66, −0.43]**	**0.000**
Gender female				0.09	[−0.04, 0.22]	0.181	0.09	[−0.04, 0.21]	0.193
SES				0.00	[−0.03, 0.04]	0.791	0.01	[−0.03, 0.04]	0.776
MVPA ≥ 5 days				**0.01**	**[−0.10, 0.12]**	0.888	0.01	[−0.10, 0.12]	0.860
Addicted × age							0.11	[−0.50, 0.72]	0.722
Problem × age							0.05	[−0.25, 0.34]	0.740
Engaged × age							0.30	[−0.21, 0.82]	0.257
Random intercept	**9.71**	**[9.63, 9.79]**	**0.000**	**9.95**	**[9.62, 10.27]**	**0.000**	**9.97**	**[9.65, 10.28]**	**0.000**

Contrast group set as reference. B (unstandardized regression coefficient) = hours. Bold writing indicates statistical significance at the *p* < 0.05 level.

**Table 4 ijerph-19-01765-t004:** Crude and adjusted associations of gaming behavior and social jetlag *.

	Model 1			Model 2			Model 3		
	b	CI 95%	*p*	b	CI 95%	*p*	b	CI 95%	*p*
Addicted	**0.35**	**[0.13, 0.57]**	**0.002**	**0.26**	**[0.02, 0.51]**	**0.038**	0.23	[−0.15, 0.61]	0.232
Problem	**0.22**	**[0.11, 0.34]**	**0.000**	**0.14**	**[0.03, 0.25]**	**0.014**	**0.18**	**[0.03, 0.33]**	**0.019**
Engaged	**0.29**	**[0.10, 0.48]**	**0.003**	**0.30**	**[0.12, 0.49]**	**0.001**	**0.40**	**[0.15, 0.65]**	**0.002**
Age 16				**0.14**	**[0.02, 0.26]**	**0.023**	**0.16**	**[0.04, 0.29]**	**0.013**
Gender female				**−0.16**	**[−0.25, −0.06]**	**0.001**	**−0.15**	**[−0.25, −0.06]**	**0.001**
SES				**0.03**	**[0.01, 0.06]**	**0.006**	**−0.03**	**[0.01, 0.06]**	**0.007**
MVPA ≥ 5 days				0.07	[−0.01, 0.16]	0.097	−0.07	[−0.01, 0.16]	0.103
Addicted × age							0.04	[−0.44, 0.53]	0.860
Problem × age							−0.08	[−0.29, 0.14]	0.475
Engaged × age							−0.16	[−0.52, 0.20]	0.385
Random intercept	**1.84**	**[1.77, 1.90]**	**0.000**	**1.52**	**[1.30, 1.74]**	**0.000**	**1.51**	**[1.29, 1.73]**	**0.000**

* Contrast group set as reference. b (unstandardized regression coefficient) = hours. Bold writing indicates statistical significance at the *p* < 0.05 level.

**Table 5 ijerph-19-01765-t005:** Crude and adjusted associations of gaming behavior and difficulties falling asleep *.

	Model 1			Model 2			Model 3		
	O.R	CI 95%	*p*	O.R	CI 95%	*p*	O.R	CI 95%	*p*
Addicted	1.48	[0.97, 2.27]	0.072	**1.89**	**[1.19, 3.02]**	**0.007**	1.83	[0.76, 4.40]	0.179
Problem	**1.58**	**[1.25, 1.99]**	**0.000**	**1.98**	**[1.51, 2.60]**	**0.000**	**2.08**	**[1.38, 3.14]**	**0.000**
Engaged	1.38	[0.96,2.00]	0.082	1.42	[0.91, 2.21]	0.120	1.30	[0.68, 2.48]	0.433
Age 16				1.17	[0.95, 1.45]	0.135	1.18	[0.94, 1.49]	0.157
Gender female				**1.71**	**[1.36, 2.15]**	**0.000**	**1.71**	[1.36, 2.15]	**0.000**
SES				0.95	[0.91, 1.00]	0.059	0.95	[0.91, 1.00]	0.057
MVPA ≥ 5 days				**1.23**	**[1.02, 1.49]**	**0.028**	**1.23**	**[1.02, 1.49]**	**0.027**
Addicted × age							1.05	[0.38, 2.93]	0.926
Problem × age							0.92	[0.55, 1.52]	0.730
Engaged × age							1.15	[0.50, 2.63]	0.743
Random intercept	**0.29**	**[0.26, 0.33]**	**0.000**	**0.23**	**[0.14, 0.36]**	**0.000**	**0.23**	**[0.14, 0.36]**	**0.000**

* Contrast group set as reference. Bold writing indicates statistical significance at the *p* < 0.05 level.

## Data Availability

The University of Bergen is the Data Bank Manager for the HBSC study. Please contact Ellen Haug for data requests.

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
