# Peer review of "Gaming Behaviors and the Association with Sleep Duration, Social Jetlag, and Difficulties Falling Asleep among Norwegian Adolescents"

_ijerph, 2022, doi:10.3390/ijerph19031765_

Round 1
Reviewer 1 Report
Please introduce “social jetlag” in a more comprehensive way, including its concept, consequences, related empirical studies, etc..
What exactly the differences of the four gaming behavior groups and how to define and measure them. It is suggested to clearly define them and introduce their related studies.
The hypotheses were not clear. How would different groups were related to sleep outcomes?
“The participants were split into four different gaming behavior groups according to which criteria they fulfilled…...” any reference to support your method of splitting the participants into these four gaming behavior groups?
The introduction did not mention physical activity or any related hypotheses. So why physical activity was measured?
A lot of important information in the section of Statistics is lacking, such as how did you deal with missing data, what kinds of data analyses were used, soft ware.
It is unclear why the authors tested the moderation effects of age. Please highlight your hypotheses and justify them in Introduction before you test them.
What are the relationships between background variables and gaming behaviors as well as sleep outcomes?
Reviewer 2 Report
See attached.

Reviewer 3 Report
Overall, the authors have explored a critical topic that is time to consider, both in terms of the increasing prevalence of problematic gaming disorder and sleep behaviour amongst adolescents and the field’s growing recognition of the importance of intervening with those addictive behaviours. I do, however, have some specific recommendations for improvement:
- Introduction: Please, provide a theoretical justification of the study.
- Method: Please, provide psychometric details about the instruments, such as reliability, validation, etc.
- Discussion:
- Please, clarify if the hypotheses of the study were satisfied.
- Please, discuss the role of the moderator variable.
- Please, suggest the theoretical and practical implications of the study.
Reviewer 4 Report
The comments and suggestions can be found in the attached file.

Round 2
Reviewer 1 Report
The authors have addressed my questions point by point and revised the manuscript accordingly. I do not have further questions on the content except I found that the some references seem problematic. For example, in line 386-388, the authors mentioned covid-19 context and cited ref 72-74. However, there was no ref 73 and 74 in the reference list and 72 was not about covid-19. I would suggest you consider cite the references below which are about gaming behaviors during covid-19. Please also double check your in-text references and reference list.
Teng, Z., Pontes, H. M., Nie, Q., Griffiths, M. D., & Guo, C. (2021). Depression and anxiety symptoms associated with internet gaming disorder before and during the COVID-19 pandemic: A longitudinal study. Journal of Behavioral Addictions, 10(1), 169-180.
Fazeli, S., Zeidi, I. M., Lin, C. Y., Namdar, P., Griffiths, M. D., Ahorsu, D. K., & Pakpour, A. H. (2020). Depression, anxiety, and stress mediate the associations between internet gaming disorder, insomnia, and quality of life during the COVID-19 outbreak. Addictive Behaviors Reports, 12, 100307.
Yang, X., Yip, B. H., LEE, K. P. E., Zhang, D. D., & Wong, S. Y. S. (2021). The relationship between technology use and problem technology use and potential psychosocial mechanisms: Population-based telephone survey in community adults during COVID-19. Frontiers in Psychology, 12, 2487.
She, R., Wong, K., Lin, J., Leung, K., Zhang, Y., & Yang, X. (2021). How COVID-19 stress related to schooling and online learning affects adolescent depression and Internet gaming disorder: Testing Conservation of Resources theory with sex difference. Journal of behavioral addictions, 10(4), 953-966.
Author Response
Dear reviewer,
Thank you for this keen and useful observation and up to date reference suggestions. The references have likely been deactivated from the reference program and misplaced in the process of editing the manuscript. They have now been changed to the original references, and we have added the suggestions. All references were checked.